# Adverse Childhood Experiences and Early Pubertal Timing Among Girls: A Meta-Analysis

**DOI:** 10.3390/ijerph16162887

**Published:** 2019-08-13

**Authors:** Lei Zhang, Dandan Zhang, Ying Sun

**Affiliations:** Department of Maternal, Child & Adolescent Health, School of Public Health, Anhui Medical University, 81th Meishan Road, Hefei 230032, Anhui Province, China

**Keywords:** abuse, pubertal timing, adversity, adverse childhood experiences, meta-analysis

## Abstract

The association between adverse childhood experiences (ACEs) and pubertal timing has been a topic of enduring controversy. A systematic search of PubMed and Web of Science databases was undertaken to quantify the magnitude of total and specific forms of ACEs effects on early pubertal timing among girls. Our search identified 3280 records, of which 43 studies with 46 independent data sets met inclusion criteria. We estimated pooled effect sizes (Cohen’s *ds*) for the association between ACEs with early pubertal timing. Total ACEs was not associated with early pubertal timing. When we examined the specific types of ACEs, associations were small to medium for father absence (*d* = −0.40, 95% confidence interval [*CI*]: −0.63, −0.16) and small for sexual abuse (*d* = −0.13, *CI*: −0.17, −0.10) and family dysfunction (*d* = −0.08, *CI*: −0.11, −0.02). We identified considerable heterogeneity between estimates for almost all of the outcomes. ACEs exposure may affect female reproductive reproduction, particularly father absence, sexual abuse, and family dysfunction. We propose that future research in this area test a theoretical model linking adversity with earlier reproductive strategy, which includes early pubertal timing as a core component linking early adversity and stress physiology with poor health outcomes later in life in females.

## 1. Introduction

The onset of puberty is pivotal for human growth and development [1]. However, among developed nations, there has been a secular trend of early pubertal timing over the past century, and in recent years, also in developing countries [2,3]. Early pubertal timing has been recognized as a risk factor for adverse health outcomes in adulthood including obesity, hyperinsulinemia, type-2 diabetes, hypertension, coronary heart disease, depression and early mortality [4,5,6,7,8]. Multiple inter-related social and environmental factors have been found to significantly influence pubertal timing, including nutrition and obesity, genetic factors, and general health status, race, and familial socioeconomic status (SES) [9].

Adverse childhood experiences (ACEs), defined as experiences that threaten the child’s bodily, familial, or social safety or security, classically include maltreatment and household dysfunction to more targeted experiences of bullying, exposure to crime, victimization, and economic disadvantage [10]. For the purposes of this meta-analysis, ACEs were defined as exposure to at least one significant form of the following childhood adversities: before the age of 18 years: physical abuse, sexual abuse, neglect (failure to provide and failure to supervise physically or emotionally), family dysfunction (parental death or serious illness, parental separation or divorce, parental incarceration, parental substance abuse, domestic violence, mental illness), low SES, and father absence (separation or divorce of the birth parents followed by absence of the birth father from the home).

ACEs are a major public health problem, with some estimates suggesting that about a third of the general population may be affected [11]. A growing number of studies have examined ACEs as risk factors for early pubertal timing, however, the association has been a topic of enduring controversy [3,9,12,13,14,15,16]. For example, Romans et al. [16] have found associations between father absence and early puberty whereas similar results were not observed by Henrichs et al. [9].

The life history theory provides a theoretical framework for explaining the individual differences in reproductive strategies from adolescence to adulthood, especially the role of family and ecological environment in regulating individual differences in puberty development and reproductive strategies. It attempts to explain the timing of reproductive development and events across the life span in terms of evolved strategies for distributing metabolic resources between the competing demands of growth, maintenance, and reproduction [17]. Specifically, there are environmental conditions in which accelerated maturational development is an adaptive response serving to maximize reproduction opportunities. Based on the life history theory, three middle-level life history models have been advanced to explain why ACEs may increase risk of early pubertal timing. First, the psychosocial acceleration theory, first advanced by Belsky, Steinberg, and Draper, 1991 postulates that ecological stressors in and around the family create conditions that undermine parental functioning and lower the quality of parental investment [18]. Under these conditions, parental investment is unreliable and/or not closely related to the reproductive success of offspring, girls are predicted to develop in a manner that accelerates pubertal maturation and sexual activity, and orients the individual towards unstable pair-bonds. Second, as a variant of the psychosocial acceleration theory, paternal investment theory parallels this logic but posits a more unique and central role for father and other men in the regulation of female pubertal timing, separate from the effects of other dimensions of psychosocial stress and support in the child’s environment [17]. The third theory is an extension on the psychosocial acceleration theory and paternal investment theory and is referred to as the child development theory. The child development theory describes how the composition and quality of family environments might influence the length of childhood as a developmental strategy. According to child development theory, the benefits of a longer childhood (delay puberty) are increased in high-quality family contexts that foster the development of sociocompetitive competencies, while the costs of cutting short childhood (accelerate puberty) are reduced in adverse family contexts that do not significantly facilitate these competencies. Children should be selected to take advantage of the benefits of high-quality parental investment and reduce the costs of low-quality parental investment by contingently altering the length of growth and development prior to reproductive maturity [17,19].

Activation of the hypothalamic-pituitary-adrenal (HPA) axis may be an underlying neuroendocrine mechanism linking ACEs and early pubertal timing, which governs both stress response and adrenarche, the first phase of pubertal maturation [20]. ACEs can alter the response of the HPA axis, with chronic stressful experiences resulting in hyperactivation followed by downregulation of the stress system. The attenuated cortisol profile may actually hasten the pubertal timing because the stress function of the HPA axis is dampened, allowing the cascade of pubertal hormones via the HPG axis to commence [21]. However, given the different forms of ACEs and the multiple indicators of HPA axis, more longitudinal studies are needed to understand the underlying mechanism better.

Variability in study design, methodology, subject characteristics, type of ACEs, and developmental timing limit the ability to draw definitive conclusions. The objective of this meta-analysis was to clarify the relationship between ACEs and early pubertal timing by means of a systematic examination of the literature, which may be beneficial for guiding future empirical and theoretical work in this area.

## 2. Materials and Methods

### 2.1. Search Strategy

The study was performed according to the recommendation of the Preferred Reporting Items for Systematic Reviews and Meta-analysis (PRISMA) guidelines [22]. A systematic database search from 1980 up to 2018 was performed on PubMed and Web of Science using the following search themes: (“ACE”, “ACEs”, “adverse childhood experiences”, “adverse childhood events”, “childhood maltreatment”, “child maltreatment”, “childhood adversity”, “child adversity”, “childhood adversities”, “child adversities”, “early adversity”, “early adversities”, “childhood physical abuse”, “child physical abuse”, “physical abuse”, “childhood sexual abuse”, “child sexual abuse”, “sexual abuse”, “childhood neglect”, “child neglect”, “childhood physical neglect”, “child physical neglect”, “physical neglect”, “child abuse”, “childhood abuse”; “child emotional abuse”, “childhood emotional abuse”, “emotional abuse”, “child emotional neglect”, “childhood emotional neglect”, “child trauma”, “childhood trauma”, “emotional neglect”, “parental incapacities”, “family incapacities”, “adverse family experiences”, “adverse family events”, “trauma”, “poverty”; “socioeconomic status”, “parental loss”, “adversity”, “early life stress”, “bullying”) combined with pubertal timing related search terms (“puberty”, “pubertal status”, “early puberty”, “early pubertal timing”, “age of menarche”, “early age of menarche”, “early maturation”, “pubertal timing”). In addition, we searched the bibliographies of the articles found on the subject [14,23]. The present analysis focused on childhood sexual abuse, physical abuse, neglect, father absence, low SES, family dysfunction. All the included studies were published in English and from peer-reviewed journals.

### 2.2. Inclusion Criteria and Exclusion Criteria

Selection criteria for the meta-analysis included articles exploring the association between ACEs and early pubertal timing. We excluded review articles, editorials, commentaries, and animal studies. The life history theory to pubertal timing pivots around the trade-off between the allocation of resources to physical growth versus production of offspring, which is particularly relevant to females. It has been applied more broadly and successfully to the question of female rather than male pubertal timing. In addition, there is a clear and easily assessed marker of female but not male pubertal timing: age at menarche [17]. Given the theoretical and empirical reasons, we included female participants only.

The majority of studies utilized age at menarche, Tanner staging scores, Pubertal Development Scale (PDS) and relative perceived timing questions to the evaluation of pubertal developmental. For studies using age at menarche, menarche occurring before the age of 11 years was considered as early pubertal timing, while after the age of 14 years as delayed pubertal timing. For those studies only reporting the correlation between ACEs with age at menarche, negative direction suggested that ACEs was associated with early pubertal timing. For studies using Tanner staging and the PDS, pubertal timing was then examined such that early pubertal timing were adolescents who scored two standard deviations above the mean and late pubertal timing were two standard deviations below the mean. Some studies also trichotomized pubertal timing groups, such that one standard deviation above the mean was categorized as early. Some studies used a single self-report item to assess relative perceived timing of puberty. Generally, this item was stated as follows “Compared with other boys/girls your age, would you say your physical development has developed ‘much earlier,’ ‘somewhat earlier,’ ‘about the same,’ ‘somewhat later,’ or ‘much later’ than other boys or girls your age?” Due to the considerable variation in the measurement method of pubertal timing, subgroup analysis was performed to examine significant sources of this heterogeneity [24].

### 2.3. Risk of Bias Assessment

Two reviewers independently assessed the methodology quality of included cohort studies using the Newcastle–Ottawa Scale (*NOS*) (range, 0 to 9 scores) [25]. Assessment items included the selection (four items, four scores), comparability (one item, two scores), and assessment of outcome (three items, three scores). Studies are then stratified by score: low quality (scored 0–3), medium quality (scored 4–6), and high quality (scored 7–9). The cross-sectional study quality was assessed using the Agency for Healthcare Research and Quality guidelines [26]. Studies are then stratified by score: low quality (scored 0–3), medium quality (scored 4–7), and high quality (scored 8–11). The third reviewer was involved when any disagreements existed. The quality scores did not affect studies for inclusion but were considered when performing sensitivity analysis and interpreting our research results.

### 2.4. Data Screening and Extraction

After duplicate citations were removed, two reviewers independently screened the titles and abstracts to identify potentially relevant citations. Then the potentially eligible studies were then screened again by full texts. The pre-designed criteria mentioned above were used to guide the entire process of screening. Subsequently, the following data were extracted from all the included studies using a pre-designed extraction form by two reviewers: (1) general information, including authors, publication year, (2) study design and specific types of ACE, (3) sample size, (4) the measure of ACEs, (5) the measure of early pubertal timing. The third reviewer was involved when any disagreements existed.

### 2.5. Effect Size Computation and Statistical Analyses

All analyses were carried out using the meta-analysis commands of Stata 11.0. For each study, we computed an effect size Cohen’s *d* [27] using the effect size calculator [28] for the association between ACEs and early pubertal timing. Cohen’s guidelines [27] were used to interpret effect sizes such that effects sizes of *ds* = 0.20 were considered small, *ds* = 0.50 were considered moderate, and *ds* = 0.80 were considered large. If correlations (r) or odds ratios (*OR*) were reported, they were converted to Cohen’s d using the formulas *d* = 2r/(1 − r^2^)^1/2^ [29] or *d* = 3^1/2^/πlog (*OR*) [30], respectively. To examine the global association between ACEs and early pubertal timing, a meta-analysis was carried out on the effects extracted from studies providing a summary measure of exposure to ACEs [31]. Unadjusted values were calculated in the program, thus avoiding biases that may have arisen from adjustments to different confounders done in each study. Heterogeneity was calculated using the *Q* test and *I*^2^ statistic and the values of *I*^2^ metric 25%, 50%, and 75% were considered as low, medium, and high heterogeneity [32]. The effect of specific types of ACEs (i.e., sexual abuse, physical abuse, neglect, father absence, low SES, family dysfunction) were tested in this meta-analysis. Due to the large overlap, these analyses were treated as independent research syntheses and no attempt was made to statistically compare these effects using meta-regression or subgroup analyses. We did sensitivity analyses by excluding outlying studies. We explored risk of publication bias using the Egger tests when sufficient studies were included in the meta-analysis (at least 10 samples).

Planned meta-regressions and subgroup analyses were conducted to explore possible sources of heterogeneity. We did univariate analyses to test the individual association of the covariates (sample size, publication year) with pooled estimates. Subgroups were delineated using the following criteria: ACEs measure (questionnaire, interview, and combination), pubertal timing measure (age at menarche, number of early menarches, PDS, Tanner stage), original variable type (categorical, continuous and others), as well as study design (cross-sectional and longitudinal).

## 3. Results

### 3.1. Search Results

As illustrated in Figure 1, after duplicates removed, the search strategy resulted in 3002 records, of which 2767 were excluded following title and abstract, and a further 193 were excluded following review of the full text, 42 studies with 45 independent data sets met full inclusion criteria. Table 1 summarizes the characteristics of the eligible studies. 26 articles used data from cross-sectional studies and 16 used data from longitudinal studies.

### 3.2. Association between Total ACEs and Early Pubertal Timing

Figure 2 shows the association between total ACEs (*n* = 35) and early pubertal timing, with an effect of Cohen’s *d*: −0.04 (95% *CI*: −0.14, 0.06). The *Q* and *I*^2^ tests indicated that the association between total ACEs and early pubertal timing was statistically heterogeneous (*I*^2^ = 99%, *p*
*<* 0.001), subgroup analyses were performed to examine significant sources of this heterogeneity. Egger’s test (*B* = 2.21, *SE* = 2.17, *p* = 0.32) showed no evidence of substantial publication bias. We examined possible changes in the effects by omitting one study at a time in the analysis but did not find any meaningful changes. Analyzing all studies except Tahirović HF et al. [61] on account of its indefinite type of ACEs, we obtained a significant results with an effect of Cohen’s *d*: −0.09 (95% *CI*: −0.17, −0.01), but the heterogeneity was still considerable (*I*^2^ = 98.3%, *p*
*<* 0.001). We excluded Alvergne et al. [36], which had a huge outlier value, from the analysis. Using the random effect model, we got a pooled Cohen’s *d*: 0.04 (95% *CI*: −0.05, 0.14), suggesting that the association between total ACEs and delay puberty was still small and not significant.

### 3.3. Associations between Specific Types of ACEs and Early Pubertal Timing

The pooled effect size of specific types of ACEs are presented in Table 2. Influence analyses were performed on each type of adversities, we did not find any meaningful changes. We found that the effect for sexual abuse (Cohen’s *d* −0.14 (95% *CI* −0.18, −0.11), *I*^2^ = 72.4, *p* < 0.001) and family dysfunction (Cohen’s *d* −0.08 (95% *CI* −0.11, −0.04), *I*^2^ = 66.9, *p* = 0.001) was small, while father absence (Cohen’s *d* −0.40 (95% *CI* −0.63, −0.16), *I*^2^ = 98.2, *p* < 0.001) was between small to medium. When we excluded Alvergne et al. [36], the effect size of father absence was reduced by nearly half (Cohen’s *d* −0.19 (95% *CI* −0.27, −0.11), *I*^2^ = 83.2, *p* < 0.001). We did not observe significant associations in other types of ACEs, or across total ACEs.

Except for physical abuse, neglect and low SES, the association between ACEs and early pubertal timing was small but significant, and the pooled effect size was highest for father absence and the least for family dysfunction. We found considerable heterogeneity between estimates for almost all of the outcomes. Heterogeneity between estimates was higher for father absence and lower for sexual abuse, relatively. Meta-regression revealed that the publication years (*p* = 0.083) and sample size (*p* = 0.661) in the primary studies did not influence the observed effect sizes.

Subgroup analyses are presented in Table 3. After subgroup analysis by ACEs measure (questionnaire, interview and combination), pubertal timing measure (age at menarche, number of early menarche, PDS, Tanner stage), original variable type (categorical, continuous and others), as well as study design (cross-sectional and longitudinal), heterogeneity between estimates is still considerable. In the subgroup analysis by pubertal timing measure, the pooled effect size was small but significant in studies measuring pubertal timing by a number of early menarche and Tanner stage. Pooled effect size of one study measuring pubertal timing by Tanner stage was Cohen’s *d*: −0.27 (95% *CI*: −0.52, −0.03) [12]. Similarly, in subgroup analysis by original variable type, the pooled effect size for categorical was small but significant.

## 4. Discussion

To our knowledge, this is the first meta-analysis to examine the association between ACEs and early pubertal timing and has extended previous studies by quantifying the effect size for early pubertal timing. Our finding suggests that sexual abuse, father absence, and growing up with challenging family circumstances were significantly associated with early pubertal timing among girls. Using Cohen’s categorization, the effect for sexual abuse and family dysfunction is small, while father absence is between small to medium. However, similar associations were not observed in other types of ACEs, or across total ACEs.

Herman-Giddens et al. [67] first reported in 1997 that child sexual abuse increased the risk of girls’ pubic hair and breast development before age 8. Subsequent studies have consistently found that sexual abuse was associated with early pubertal timing [16,42,43]. Recently, a cohort study of the Avon Longitudinal Study of Parents and Children (ALSPAC) mothers showing that total ACEs was not associated with age at menarche, while childhood sexual abuse was associated with lower age at menarche, which is consistent with our findings [54]. Sexual abuse is often regarded as the most prevalent health problem children face with the most serious array of consequences, while in our study, early pubertal timing may be more affected by father absence instead of sexual abuse.

As early as the 1980s to the 1990s, Draper and Harpending [68] and Belsky et al. [18] began to focus on the association between father absence and reproductive behaviors in adulthood, which has received increased attention over recent years. Majority of these studies have indicated a significant association between father absence and early pubertal timing. A study from a large UK birth cohort (ALSPAC) demonstrated that girls from father-absent homes had earlier menarche than girls from father-present homes [46]. The observed association between father absence and early pubertal timing has also been supported by other articles [20,23]. Moreover, the earlier that father absence occurs, the earlier girls tend to experience puberty, there is evidence to suggest that exposure to father absence during the first five years of life is more strongly associated with early menarche than father absence later in childhood [24].

It has been previously argued that the effects of age at menarche may be explained by low SES arising as a result of father absence, for example, decrease in family income, major financial problems [69]. However, in this meta-analysis, we did not find a significant association between low SES and early pubertal timing. When we excluded studies in which pubertal timing measured by Tanner stage or PDS [12,15,23,60], results suggested that girls living in low SES households may experience delayed onset of menarche (Cohen’s *d*: 0.12, 95% CI: 0.003, 0.25). A longitudinal study on 1091 black and 986 white girls from the three sites in the United States found that high SES was a protective factor for the risk of early menarche among white girls but opposite among black girls [14]. Given the challenges in methodology and definition of SES along with socially heterogeneous of study populations, it is difficult to draw firm conclusions concerning the relationship between low SES and early pubertal timing.

The association between family dysfunction and early pubertal timing was weak, we suspect that family dysfunction may be a proxy variable that represents a range of other family characteristics (including parental divorce, separation, mental illness, and family conflict) and have an effect similar to father absence. However, when we excluded three articles involving parental divorce or separation, the effect size was small but still significant. The effect of total ACEs on adolescent development was not significant. One explanation may be that there are great individual differences in the types, timing, duration, and severity of ACEs. Another explanation is that the association may differ depending on race/ethnicity. As demonstrated in prior research, high SES was a protective factor for the risk of early menarche among white girls but opposite among black girls [14].

Overall, despite some inconsistencies across specific forms of ACEs with early pubertal timing, these results support that ACEs exposure may affect female reproductive function. Given the association between ACEs and early pubertal timing, clinicians should sensitively enquire about past childhood experiences, and tailor support accordingly, thus resetting their trajectory for lifelong health and reducing the risk of chronic diseases and conditions.

There are also several limitations. Firstly, most of the measurements relied on retrospective self-report which is subject to recall or reporting biases. Secondly, this sample was restricted to females, and these findings cannot be extrapolated to males. In addition, the analysis revealed substantial statistical heterogeneity, which allows for less confidence in the estimated effect sizes, but is not surprising given the methodological and analytic variances in the identified studies. We cannot rule out the effect of interactions of ACEs with other factors (parental early puberty history, physical activity, etc.) in the current meta-analysis because most studies did not adjust for these interactions, or partly adjusted for these factors as possible moderators. The current meta-analysis focused on specific types of ACEs (sexual abuse, physical abuse, neglect, father absence, low SES, and family dysfunction). Nevertheless, adversity is a heterogeneous concept (including types of ACEs not considered here, such as bullying, exposure to war, and natural disasters, etc.). Although most studies measured ACEs at any point in childhood or adolescence, we were unable to obtain sufficient information about the frequency, timing, and duration of ACEs. Additionally, studies without sufficient statistical information for the computation of effects were not included in our analysis. Nevertheless, the thorough search of literature ensures a full coverage of the existing evidence and makes the results more convincing. Moreover, the large sample of the study population derived from the included studies enhances the possibility of effect detection with reasonable statistical power.

These limitations also suggest some directions for future research. Identification of distinct dimensions, duration and sensitive periods of adverse childhood experience that differentially influence pubertal development are required to uncover mechanisms that explain how childhood adversity is associated with female reproductive strategies.

## 5. Conclusions

In summary, our analyses indicate that father absence is more strongly related to early pubertal timing among girls, followed by sexual abuse and family dysfunction. Additional research is needed to better understand the mechanisms driving these relationships. Due to the high heterogeneity, a conclusive statement on the effect size of the association between ACEs and early pubertal timing should be made with caution.

## Figures and Tables

**Figure 1 ijerph-16-02887-f001:**
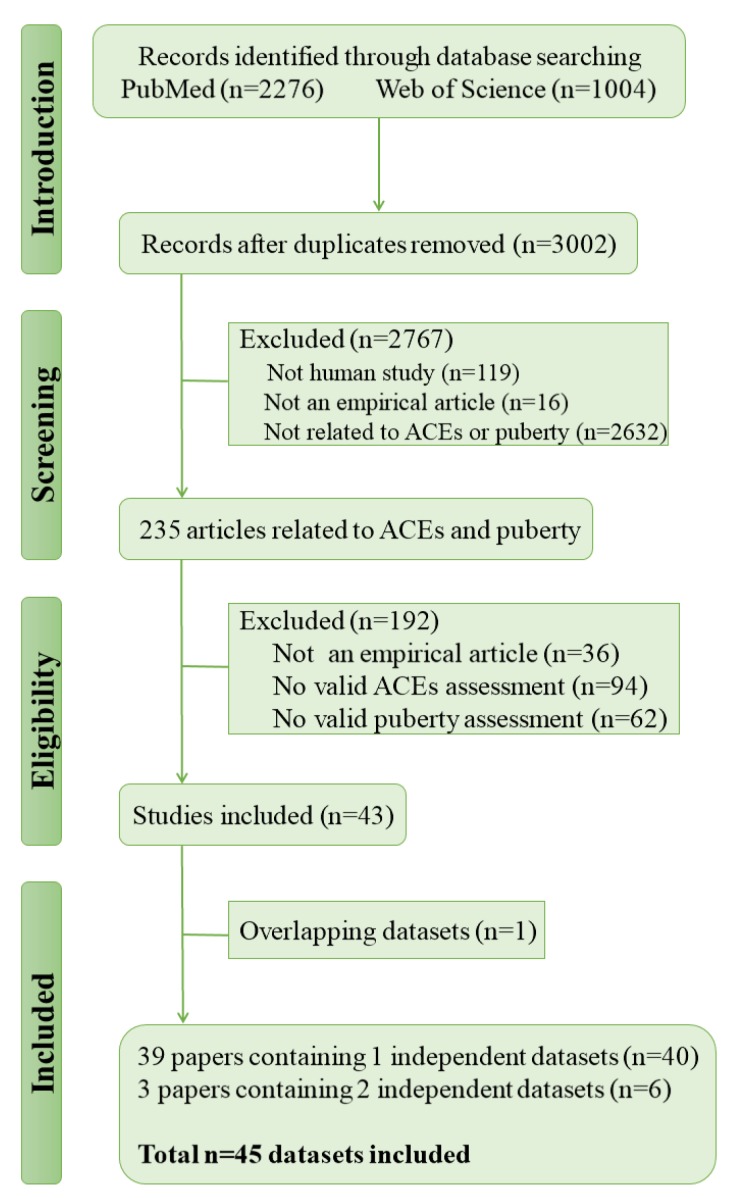
Flow chart of study selection.

**Figure 2 ijerph-16-02887-f002:**
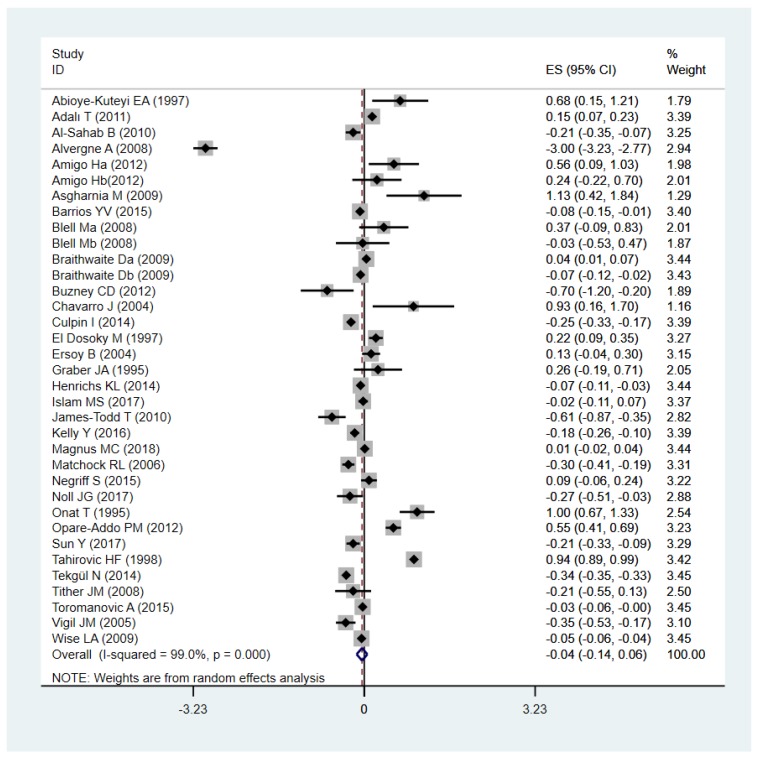
Forest plot of effect sizes reported as Cohen’s *d* (x-axis) evaluating adverse childhood experiences (ACEs) and early pubertal timing using the random effects model.

**Table 1 ijerph-16-02887-t001:** Descriptive statistics and study characteristics for included studies.

Study	Study Design	N	Age (Year)	ACEs Type	ACEs Measure	Puberty Measure
Abioye-Kuteyi EA (1997) [33]	C	60	14.2	low SES	questionnaire	AAM
Adalı T(2011) [34]	C	2789	15–49	low SES	combination	AAM
Al-Sahab B (2010) [35]	C	1403	14–17	low SES	questionnaire	NO.
Alvergne A (2008) [36]	C	708	20.9	FB	questionnaire	AAM
Amigo H (a) (2012) [37]	C	127	8–16	low SES	interview	AAM
Amigo H (b) (2012) [37]	C	114	8–16	low SES	interview	AAM
Asgharnia M (2009) [38]	C	91	11–16	low SES	combination	AAM
Barrios YV (2015) [39]	C	1499	28	PA, SA	questionnaire	NO.
Bleil ME (2013) [40]	C	650	34.9	FD	combination	AAM
Blell M (a) (2008) [41]	L	94	49–51	low SES	questionnaire	AAM
Blell M (b) (2008) [41]	L	106	49–51	low SES	questionnaire	AAM
Boynton-Jarrett R (2012) [42]	L	4524	16	PA, SA, neglect	combination	NO.
Boynton-Jarrett R (2013) [43]	L	67,658	25–44	abuse	questionnaire	NO.
Braithwaite D (a) (2009) [14]	L	1091	18–19	low SES	interview	NO.
Braithwaite D (b) (2009) [14]	L	986	18–19	low SES	interview	NO.
Buzney CD (2012) [44]	C	70	19–25	low SES	combination	AAM
Chavarro J (2004) [45]	C	30	15−42	low SES	questionnaire	AAM
Culpin I (2014) [46]	L	3785	8–17	FB	questionnaire	AAM
Deardorff J (2011) [23]	L	444	8–10	FB, low SES	interview	Tanner staging
Junqueira DLM (2003) [47]	C	2053	9–19	low SES	questionnaire	AAM
El Dosoky M (1997) [48]	C	929	9–18	low SES	questionnaire	AAM
Ersoy B (2004) [49]	C	534	15.7	low SES	combination	AAM
Graber JA (1995) [50]	L	75	11.93	FB, low SES	questionnaire	AAM
Henrichs KL (2014) [9]	C	3288	45.7	PA, SA, neglect, low SES, FB, FD	interview	NO.
Islam MS (2017) [3]	C	680	14	low SES	questionnaire	NO.
James-Todd T (2010) [51]	L	237	—^a^	low SES	questionnaire	AAM
Jorm AF (2004) [52]	C	3702	—^a^	PA, SA, neglect, FB	questionnaire	AAM
Kelly Y (2016) [53]	L	5839	11.2	low SES	interview	PDS
Magnus MC (2018) [54]	L	8984	28.5	Total adversity, SA, FD	questionnaire	AAM
Matchock RL (2006) [55]	C	1896	20	FB	questionnaire	AAM
Mendle J (2006) [56]	C	1284	24.5	FB, FD	interview	AAM
Mendle J (2016) [20]	L	6273	28.7	SA, PA, neglect	interview	AAM
Moffitt TE (1992) [57]	L	416	15	FB, low SES, FD	combination	AAM
Negriff S (2015) [15]	L	213	8–13	SA	questionnaire	PDS
Noll JG (2017) [12]	L	173	6–16	SA	interview	Tanner staging
Onat T (1995) [58]	L	169	8.5–13.4	low SES	questionnaire	AAM
Opare-Addo PM (2012) [59]	C	720	7–17	low SES	questionnaire	AAM
Romans SE (2003) [16]	C	488	39.1	low SES, PA, SA, FB	interview	NO.
Sun Y(2017) [60]	L	1770	10–11	low SES	questionnaires	PDS
Tahirovic HF (1998) [61]	C	6077	8–17	War	questionnaire	AAM
Tekgül N (2014) [62]	C	61,293	15–49	low SES	interview	AAM
Tither JM (2008) [63]	C	136	16–44	FB	questionnaire	AAM
Toromanović A (2015) [64]	C	22,469	9–17.5	FD	questionnaire	AAM
Vigil JM (2005) [65]	C	616	26.9	SA	questionnaire	AAM
Wise LA (2009) [66]	C	35,330	M = 38	PA, SA	questionnaire	NO.

L = longitudinal, C = cross-sectional, PA = physical abuse, SA = sexual abuse, FB = father absence, FD = family dysfunction, SES = socioeconomic status, AAM = age at menarche, NO. = number of participants with early menarche, PDS = Pubertal Development Scale. ^a^ Dashed cells indicate insufficient information available in study.

**Table 2 ijerph-16-02887-t002:** Pooled effect size of specific types of ACEs.

Types of Adversity	*K*	Cohen’s *d* (95% *CI*)	*I*^2^ (%), *p* Value
Sexual abuse	12	−0.14(−0.18, −0.11)	72.4, <0.001
Physical abuse	8	−0.03 (−0.07,0.01)	91.5, <0.001
Neglect	4	0.02 (−0.1,0.14)	72.8, 0.011
Low SES	25	0.07 (−0.03,0.18)	97.4, <0.001
Father absence	12	−0.40 (−0.63, −0.16)	98.2, <0.001
Family dysfunction	11	−0.08 (−0.11,−0.04)	66.9, 0.001

Data in parentheses are 95% Cis, ES = effect size (Cohen’s *d*), SES = socioeconomic status.

**Table 3 ijerph-16-02887-t003:** Subgroup analyses of Associations between ACEs and Early Pubertal Timing.

Moderator	Cohen’s *d*	95% Confidence Interval		*p-*Value	*I*^2^ (%)
		Lower	Upper		
**ACEs measure**					
questionnaire	−0.06	−0.22	0.10	0.493	99.1
interview	−0.07	−0.23	0.09	0.382	98.7
combination	0.03	−0.14	0.20	0.728	86.3
**Puberty measure**					
No. of early menarche	−0.050	−0.090	−0.011	0.012	81.3
PDS	−0.109	−0.268	0.050	0.180	82.1
Tanner staging	−0.270	−0.515	−0.025	0.031	—
age at menarche	0.013	−0.172	0.199	0.889	99.3
**Original variable type**					
categorical	−0.073	−0.114	−0.031	0.001	82.1
continuous	−0.011	−0.254	0.233	0.931	99.5
others	0.098	−0.137	0.333	0.415	65.3
**Study design**					
cross-sectional	−0.050	−0.194	0.094	0.492	99.3
longitudinal	−0.051	−0.141	0.038	0.262	91.4

Cohen’s *d* effect sizes coded such that more negative values reflect a stronger association between early pubertal timing and ACEs. No. of early menarche = number of participants with early menarche.

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
