# Peer review of "Adverse Childhood Experiences and Early Pubertal Timing Among Girls: A Meta-Analysis"

_ijerph, 2019, doi:10.3390/ijerph16162887_

Round 1

Reviewer 1 Report

Thank you for conducting the meta-analysis on the relationship between ACEs and early pubertal timing. My suggestions for improving the manuscript follow:

Title: change to "Adverse childhood experiences and early pubertal timing among girls: A meta-analysis"

Abstract: Second and third sentences are incomplete; combine. Move sentence (line 16; "We estimated pooled effect sizes") and insert before sentence (line 18; "Total ACES was not associated")

Introduction: Line 39, insert "and" to maltreatment and household dysfunction. Line 57-59, confusing sentence, please revise and changing maximizing to maximize

Methods: Line 125, change "occurred" to "occurring". Line 126, insert "studies" to "For those studies only reportING (not reported). Line 180, invert labels underneath Figure 1. 

Results: Table 1 is outstanding! Line 195, change "showed" to "shows" the association. Line 232, add "is" to between estimates is still considerable."

Discussion: Line 251, the phrase "sexual abuse may be the most common and severe one we can capture" is confusing. Can you clear it up? Line 255, consider deleting the phrase "a key component in ACES," Line 272, add "s" to "white girls" Line 279, change "involved to" to "involving parental divorce" Line 281-282, revise the last half of the sentence, and offer tentative explanation for why there was no effect for total ACEs. Line 290, change "disadvantages" to "limitations" (I encourage you to balance this out with the strengths of your work, especially the sample size and number of studies). Line 290, change "relies" to "rely" or "relied." Line 297, move "partly" to "partly adjusted" Line 309, use "among" instead of "in" (later in life among females). Line 311, use "is" instead of "are" (timing is mostly affected). 

Reviewer 2 Report

This is a very interesting paper that was carefully done and attempts to aggregate studies on the relationship between pubertal timing and abuse, neglect, father absence and family environmental issues for females.  The findings support some previous work and indicate significant pathways for future studies. Other than problems with English, the paper is ready for publication.
